# Determinants of Antibody Response to a Third SARS-CoV-2 mRNA Vaccine Dose in Solid Organ Transplant Recipients: Results from the Prospective Cohort Study COVAC-Tx

**DOI:** 10.3390/vaccines10040565

**Published:** 2022-04-06

**Authors:** Daniel Balsby, Anna Christine Nilsson, Sören Möller, Susan Olaf Lindvig, Jesper Rømhild Davidsen, Rozeta Abazi, Mikael Kjær Poulsen, Inge Kristine Holden, Ulrik Stenz Justesen, Claus Bistrup, Isik Somuncu Johansen

**Affiliations:** 1Department of Infectious Diseases, Odense University Hospital, 5000 Odense, Denmark; daniel.balsby@rsyd.dk (D.B.); susan.olaf.lindvig@rsyd.dk (S.O.L.); inge.holden@rsyd.dk (I.K.H.); 2Department of Clinical Research, University of Southern Denmark, 5000 Odense, Denmark; christine.nilsson@rsyd.dk (A.C.N.); soren.moller@rsyd.dk (S.M.); jesper.roemhild.davidsen@rsyd.dk (J.R.D.); ulrik.stenz.justesen@rsyd.dk (U.S.J.); claus.bistrup@rsyd.dk (C.B.); 3Department of Clinical Immunology, Odense University Hospital, 5000 Odense, Denmark; 4Open Patient Data Explorative Network, Odense University Hospital, 5000 Odense, Denmark; 5Department of Respiratory Medicine, Odense University Hospital, 5000 Odense, Denmark; 6Department of Gastroenterology, Odense University Hospital, 5000 Odense, Denmark; rozeta.abazi@rsyd.dk; 7Department of Cardiology, Odense University Hospital, 5000 Odense, Denmark; mikael.kjaer.poulsen1@rsyd.dk; 8Department of Clinical Microbiology, Odense University Hospital, 5000 Odense, Denmark; 9Department of Nephrology, Odense University Hospital, 5000 Odense, Denmark

**Keywords:** SOT, humoral response, third dose, COVID-19, vaccine

## Abstract

Background: We studied factors related to humoral response in solid organ transplant (SOT) recipients following a three-dose regimen of an mRNA-based SARS-CoV-2 vaccine. Method: This was a prospective study of SOT recipients who received a third homologous dose of the BNT162b2 (Pfizer–BioNTech) vaccine. The anti-spike S1 IgG response was measured using the SARS-CoV-2 IgG II Quant assay (Abbott Laboratories) with a cut-off of 7.1 BAU/mL. Multiple logistic regression was used to determine the factors associated with humoral response. Results: In total, 395 SOT recipients were included. Anti-spike IgG was detected in 195/395 (49.4%) patients after the second dose and 261/335 (77.9%) patients after the third dose. The overall mean increase in antibody concentration after the third dose was 831.0 BAU/mL (95% confidence interval (CI) 687.4–974.5) and 159 (47.5%) participants had at least a 10-fold increase in antibody concentration after the third dose. The increase in antibody concentration was significantly higher among patients with detectable antibodies after the second dose than those without. Cumulative time from transplantation and liver recipients was positively associated with an antibody response, whereas older age, administration of prednisolone, and proliferation inhibitors were associated with diminished antibody response. Conclusion: Although the third dose of the BNT162b2 vaccine improved humoral responses among SOT non-responders following the second dose, the overall response remained low, and 22.1% did not develop any response. Patients at risk of a diminished vaccine response require repeated booster doses and alternative treatment approaches.

## 1. Introduction

As a result of lifelong immunosuppressive treatment, solid organ transplant (SOT) recipients are at a higher risk of contracting severe infections, including coronavirus disease 2019 (COVID-19) [1,2].

In contrast to the general population, SOT recipients have shown suboptimal immunogenicity from COVID-19 vaccines [3,4,5,6,7,8,9,10,11,12]. This blunted response is correlated with several patient- and treatment-related risk factors that have not been fully elucidated.

New and expensive treatment modalities such as monoclonal antibodies, which are only indicated in newly infected high-risk patients or as pre-exposure prophylaxis, have emerged. Thus, to prioritize these expensive treatments to those with a poor vaccine response, it is important to identify the risk factors among those with diminished vaccine response.

Information regarding the frequency with which SOT recipients should receive booster doses is still not clear, but would be of great importance for obtaining optimal immunogenicity in this vulnerable group [13,14]. To investigate this, we studied the humoral immune response in SOT recipients following a three-dose regimen of mRNA-based SARS-CoV-2 vaccine.

## 2. Method

Since 29 January 2021, all SOT recipients (≥18 years of age) from Southern Denmark were invited to participate in the study (Danish Ethical Committee, record No. 77786).

All participants received vaccination as part of the national COVID-19 vaccination program, and a third dose was offered to prioritized target groups, including SOT recipients, in September 2021. The study population has been described previously [6], and patients who completed the two-dose SARS-CoV-2 mRNA vaccine series were included and followed through to 4 January 2022. As per the protocol, blood samples were drawn 4, 8, and 20 weeks after the second and third vaccinations.

The SARS-CoV-2 spike S1 IgG response was measured using the SARS-CoV-2 IgG II Quant Assay (Abbott Laboratories). The relationship between the Abbott arbitrary units (AU)/mL unit and the World Health Organization BAU/mL unit follows the equation BAU/mL = 0.142 × AU/mL, corresponding to a cut-off of 7.1 BAU/mL.

Participants treated with monoclonal antibodies before the scheduled samples were excluded from further analyses.

Categorical data are described as counts and proportions. Group comparisons were performed using the Fisher’s exact test. Continuous variables were described as medians with interquartile ranges (IQRs) and compared using the Wilcoxon rank-sum test. Changes in antibody concentrations between time points were estimated using mixed-effects regression models and reported as mean differences with 95% confidence intervals (CI). Associations between variables of interest and the defined antibody response as outcomes were investigated using multiple logistic regression analysis. Statistical significance was set at *p* < 0.05.

## 3. Results

Serum was obtained from 395 recipients after the second dose and 335 after the third dose. The general characteristics of the study population are summarized in Table 1. The median age of the cohort was 58.2 years (IQR, 48.0–66.9), with 58.5% being male. Most SOT recipients were kidney transplant recipients (73.2%). Two patients were previously diagnosed with COVID-19 before the first dose. All participants received the BNT162b2 vaccine (Pfizer, BioNTech). The median time from the second to the third dose was 231 days (IQR 201–241), and the median time from the second and third doses to blood sampling was 72 days (IQR 56–80) and 40 days (IQR 33–52), respectively. SARS-CoV-2 spike S1 IgG antibodies were detected in 195 of 395 patients (49.4%) after the second dose and 261 of 335 patients (77.9%) after the third dose.

Serum concentrations of antibodies after the second and third vaccinations in the five SOT groups are depicted in Figure 1a,b. The overall mean increase in antibody concentrations after the third dose was 831.0 BAU/mL (95% CI 687.4–974.5). In total, 159 participants (47.5%) showed at least a 10-fold increase in antibody concentrations following the third dose of the vaccine. The increase in antibody concentration was significantly higher among patients with detectable antibodies after the second dose than among those without (mean increase 1371.9 BAU/mL (95% CI 1185.8–1558.1) versus 293.2 BAU/mL (95% CI 107.63–478.8), *p* < 0.001).

Among the 200 patients who were seronegative after the second dose, 95 (47.5%) became seropositive after the third dose, while 73 (36.5%) remained negative (missing data on 32 patients (16.0%)). Among the responders after the second dose, 1 (0.5%) waned immunity, while 166 (85.1%) remained seropositive (missing data on 28 patients (14.4%)).

In our multiple regression analysis, we found that a longer time since transplantation (OR 1.16, 95% CI 1.03–1.31, *p* = 0.013) and liver recipients (OR 8.27, 95% CI 2.24–30.46, *p* = 0.002) were positively associated with an antibody response, whereas increased age, administration of prednisolone, and especially proliferation inhibitors were associated with a reduced antibody response (OR 0.07, 95% CI 0.01–0.69, *p* = 0.023) (Table 2). A shorter time between vaccinations was not significantly associated with a positive antibody response (229 days (IQR 198 to 241) vs. 233 days (IQR 211 to 240), *p* = 0.981) for responders vs. non-responders, respectively.

A total of 35 patients (8.9%) developed breakthrough COVID-19 infection (3 after the second dose and 32 after the third dose). The median antibody titers were 12.2 BAU/mL (IQR 7.1–465.3) (two participants had no antibody results prior to infection). Fourteen participants (42.4%) were non-responders and seven responders (21.2%) had anti-SARS-CoV-2 antibody titers above 1000 BAU/mL prior the infection. The median time from antibody testing to breakthrough infection was 38 days. Sixteen patients with mild disease were treated with monoclonal antibodies and three were hospitalized (one patient with a positive response to all doses). 

## 4. Discussion

This study showed that a third dose of the mRNA vaccine in SOT recipients resulted in a significant increase in antibody titers. Furthermore, the proportion of seronegative patients has decreased. SARS-CoV-2 spike IgG antibodies were detected in 49.4% of SOT recipients after the second dose and 77.9% after the third dose. The overall increase in antibody titers after the third dose was the highest among those with detectable antibodies after the second dose. Furthermore, antibody titers did not decrease significantly until the third vaccine. The results of this study are similar to those of previous studies of SOT recipients [11,12,15].

Protective levels of anti-SARS-CoV-2 antibodies after infection or vaccination have not yet been established. In our study population, the median antibody concentration after the second dose was 7.1 (IQR 7.1–73.2) and increased to 319.5 (IQR 17.2–1244.7) following the third dose. This was in line with the findings reported among non-immunocompromised adults aged ≥60 years by Eliakim-Raz et al. [16]. The authors reported the median antibody concentration level increased significantly after the third dose, from a median of 440 AU/mL (IQR, 294–923) to 25 468 AU/mL (IQR, 14 203–36 618) and all participants became seropositive. However, this increase in antibody concentration was limited in the current study and 22.1% of the patients had no humoral response after the third dose. Our multivariate analysis identified increased age, shorter time since transplantation and treatment with prednisolone and proliferation inhibitors to be associated with a diminished antibody response. 

As in many other countries, the fourth dose has only been recommended to prioritize target groups, including SOT recipients. The first publications about a fourth dose have been published, and the results showed that repeated boosters elicited a higher response in non-responders or increased the level of antibody concentrations [17]. Even data following a fifth vaccination dose have just been published, showing that suboptimal responders to the fourth dose could benefit from the fifth dose [18]. However, more data are needed for immunization strategies for prioritized target groups. 

The number of vaccine-breakthrough infections was relatively low (8.9%), which was lower than the vaccinated background population (8.9% vs. 15.6%) [19]. The majority of these SOT recipients had mild infection, only three were admitted to the hospital, and none required intensive care treatment. 

The limitations of this study were missing antibody values after the third dose and the lack of a standardized time period for spike IgG measurements between vaccines. These limitations may have underestimated the proportion of responses and the level of IgG concentrations between doses. Furthermore, the number of heart and lung transplant recipients is low. Further limitations were the lack of a neutralizing assay, although the detection of spike IgG has been correlated with the presence of neutralizing antibodies and data on cellular response [20].

In conclusion, among SOT patients, a third dose of an mRNA-based SARS-CoV-2 vaccine improved the proportion of humoral responses among non-responders following the second dose. However, the overall antibody response rate was low. The vaccine response was negatively affected by increased age, shorter time since transplantation, and administration of prednisolone and proliferation inhibitors. Alternative treatment approaches such as pre-exposure prophylaxis for those at risk for diminished vaccine response, repeated booster doses and treatment with monoclonal antibodies are needed. 

## Figures and Tables

**Figure 1 vaccines-10-00565-f001:**
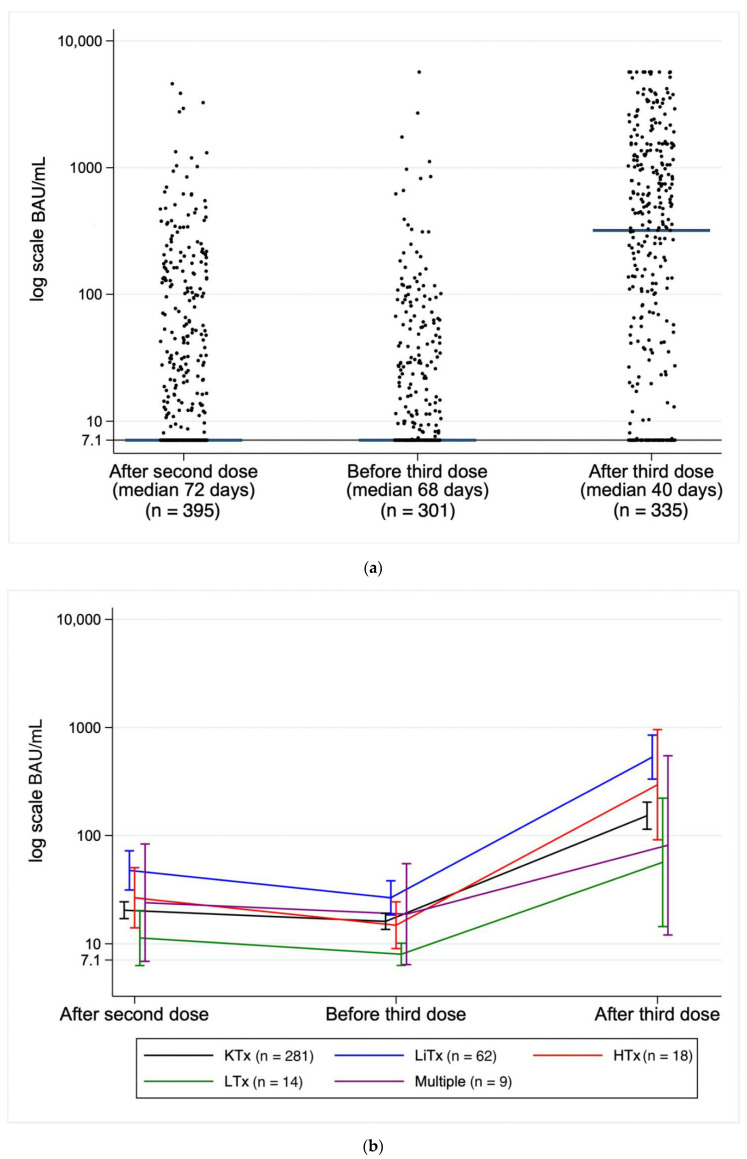
(**a**): Anti-SARS-CoV-2 antibody after the second and the third dose of a SARS-CoV-2 mRNA vaccine among solid organ transplant recipients, depicted on a log-scale. Median after second dose: 7.1 (IQR 7.1–73.2)*;* Median after third dose: 319.5 (IQR 17.2–1244.7). (**b**): Anti-SARS-CoV-2 antibody titers to SARS-CoV-2 vaccine correlated to transplant. Transplants: KTx (kidney), LiTx (liver), HTX (heart), LTx (lung), Multiple. From after the second dose to before third dose mean increase/decrease: −74.4 BAU/mL (95% CI: −208.9–60.2). From after the second dose to after the third dose mean increase/decrease: +831.2 BAU/mL (95% CI: 705.8–956.6).

**Table 1 vaccines-10-00565-t001:** Characteristics of solid organ transplant recipients who completed the third dose of a SARS-CoV-2 mRNA vaccine.

Characteristics	Total Cohort after Second Dose	Responders after Third Dose	Non-Responders after Third Dose	*p*-Value
N	395	261 (77.9%)	74 (22.1%)	
Antibody concentration (median)	7.1 (IQR 7.1–73.2)	605.5 (IQR 157.3–1543.6)	<7.1	<0.001
Age (median)	58.2 (IQR 48.0–66.9)	58.1 (IQR 48.7–66.4)	60.2 (IQR 54.4–71.1)	0.036
Sex (males)	225 (58.4%)	152 (58.9%)	42 (56.8%)	0.789
BMI	26.0 (IQR 23.0–29.1)	26.0 (IQR 23.1–28.9)	25.9 (IQR 21.9–29.7)	0.544
Time from Tx ^b^ to vaccine months(median)	86.7 (IQR 50.3–159.3)	95.5 (IQR 55.7–183.1)	51.8 (IQR 25.9–127.9)	<0.001
Organ transplant				<0.001
Kidney	281 (73.2%)	182 (70.5%)	58 (78.4%)	
Liver	62 (16.2%)	54 (20.9%)	3 (4.1%)	
Heart	18 (4.7%)	12 (4.7%)	3 (4.1%)	
Lung	14 (3.7%)	6 (2.3%)	6 (8.1%)	
Combined ^a^	9 (2.3%)	4 (1.6%)	4 (5.4%)	
Immunosuppressive treatment				
Prednisolone	95 (24.0%)	55 (21.1%)	26 (35.1%)	0.020
CNI ^c^	311 (78.7%)	198 (75.9%)	61 (82.4%)	0.273
Proliferation inhibitor ^d^	365 (92.4%)	237 (90.8%)	73 (98.7%)	0.022
mTOR inhibitor ^e^	1 (0.3%)	1 (0.4%)	0 (0%)	1.000

Note: ^a^ Kidney/liver, kidney/heart, and heart/lung. ^b^ Transplantation. ^c^ Calcineurin inhibitor (CNI): tacrolimus and cyclosporine. ^d^ Proliferation inhibitors: mycophenolate and azathioprine. ^e^ Mammalian target of rapamycin inhibitor (mTORi): sirolimus or everolimus.

**Table 2 vaccines-10-00565-t002:** Association between patient characteristics and immunosuppression following the third dose of a SARS-CoV-2 mRNA vaccine among solid organ transplant recipients.

Responders	Odds Ratio	*p*	[95% CI]
Age	0.97	0.017	0.95–0.99
Female	0.87	0.629	0.48–1.56
BMI	1.00	0.954	0.94–1.06
Month from Tx	1.16	0.013	1.03–1.31
SOT ^a^			
Kidney	1 (ref.)		
Liver	8.27	0.002	2.24–30.46
Heart	0.50	0.433	0.09–2.85
Lung	0.81	0.758	0.20–3.20
Combined	0.42	0.307	0.08–2.23
Immunosuppressive treatment			
Prednisone	0.31	0.002	0.15–0.66
CNI	1.03	0.938	0.47–2.29
Proliferation inhibitor	0.07	0.023	0.01–0.69
mTOR inhib. ^a^	1		

^a^ (mTor inhibitor): too few observations for multivariate analysis.

## Data Availability

The data presented in this study are available on request from the corresponding author.

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
