# Peer review of "Determinants of Antibody Response to a Third SARS-CoV-2 mRNA Vaccine Dose in Solid Organ Transplant Recipients: Results from the Prospective Cohort Study COVAC-Tx"

_vaccines, 2022, doi:10.3390/vaccines10040565_

Round 1

Reviewer 1 Report

Review of the MS #1651118 submitted by Balsby et al. for publication in Vaccines.

The authors tested sera from 395 solid organ transplant (SOT) recipients who had received three doses of a SARS-CoV-2 mRNA vaccine for anti-SARS-CoV-2 IgG. Sera were collected after the second and third doses and tested with a standardized IgG assay based on viral receptor binding domain (RBD) as antigen. The authors found that only half of the patients developed anti-RBD IgG after the second dose. After the third dose, the prevalence of anti-RBD IgG increased to 78%. Accordingly, an increase in the mean anti-RBD IgG concentration was also observed. The authors noted that the increase was higher in patients who had already developed anti-RBD IgG after the second vaccination. In addition, several factors (liver transplantation, longer time after transplantation) were positively associated with the success of anti-RBD IgG development after three doses of vaccination, whereas others (older age and use of corticosteroids, mycophenolate, and azathioprine) were negatively associated.

The manuscript is of interest and relevance. Although the problem of low immunogenicity of vaccines used in SARS-CoV-2 recipients is well known, there are few studies on the association with specific patients or treatment factors.

Major points:

1) Can the authors provide information on what anti-RBD IgG concentrations can be expected in healthy individuals shortly after the second or third vaccination? The test used is very well standardized and has already been applied in several studies. Therefore, the data should be available. This would facilitate comparison with the results presented here.

2) Almost a quarter of patients do not develop sufficient (?) anti-RBD IgG antibodies even after a third vaccination. However, it is encouraging to read that the proportion of (severe) breakthrough infections is nevertheless quite low. This is remarkable, as especially in Denmark a very high SARS-CoV-2 prevalence has been observed recently.

Have other authors also reported a comparatively low rate of breakthrough infections in triple-vaccinated SOT recipients?

It would be interesting to know which viral variant has been detected in the 35 breakthrough infections. Is there any data on this?  

Relatively high anti-RBD IgG concentrations >1000 IU/ml were measured in seven patients with breakthrough infection. Did this measurement take place before or after infection? In other words, are these values possibly influenced by the infection?

Furthermore, the question arises to what extent other factors also play a role here. In my opinion, the development of cellular immunity could also have been investigated. For example, it would have been interesting to see to what extent virus-specific T cells were detectable in the non-responders. For this purpose, there are now relatively easy-to-perform tests for measuring the interferon gamma release. The authors point out some of the limitations (only one IgG test, no consideration of neutralizing antibodies, no consideration of cellular immunity) themselves.

Minor points:

1) Instead of titers, it is better to speak of IgG concentrations.

2) The indication "BAU/ml" should be used consistently in the text and in the figures. Currently, "BAU/ml" and "IU/ml" are used.

3) Figure 1a lacks information on the number of cases examined and the 95% confidence interval. It may also be possible to add the mean time interval (in days) between vaccination and blood collection. Likewise, the median concentration after the third vaccination (blue line) should be added.

4) In Figure 1b, the scaling of the ordinate should be changed to allow comparability with Figure 1a. Why are there no values >1000 IU/ml in this figure, as in Figure 1a? Similarly, the number of cases and the mean time interval (in days) between vaccination and blood collection should be reported. Here, data before the third vaccination can be found, which are missing in Figure 1a.

Author Response

We thank the reviewer for the comments and suggestions. We agree with the reviewer and we revised the manuscript according to the suggestions.

Major points:

  1. We thank the reviewer for this comment and we have now added following clarification and reference in the discussion:

Protective levels of anti-SARS-CoV-2 antibodies after infection or vaccination have not yet been established. In our study population, the median concentrations after second dose was 7.1 (IQR 7.1-73.2) and increased to 319.5 (IQR 17.2-1244.7). This was in line with the findings reported among non-immunocompromised adults aged ≥60 years by Eliakim-Raz et al. The authors reported  the median antibody concentration level increased significantly after the third dose, from a median of 440 AU/mL (IQR, 294-923) to 25 468 AU/mL (IQR, 14 203-36 618)  and all participants became seropositive. However, this increase in antibody concentration was limited in the current study and 22.1% of the patients had no humoral response after the third dose. 

  1. Luckily, the breakthrough infections have been not severe in our population. As we mentioned in the manuscript, the majority of the breakthrough infections have been seen after the third vaccine and these were in a period the Omicron variant was dominant in Denmark. This is one explanation for the majority of the SOT recipients had mild disease, since this is characteristic for the Omicron variant. Unfortunately, we do not have access to variant data.

The other reason for the low breakthrough infection rate in this group of patient is that use of personal protective measurements have been very common in Denmark.

In studies, which has reported breakthrough infections, the immunocompromised population had a higher risk of breakthrough infection. However, the comparison is not straightforward since the methods are not comparable.       

Minor points:

  1. We thank the reviewer for this suggestion and we have now used concentration instead for titers.
  2. We agree with the reviewer and we have used BAU/ml throughout the paper.
  3. We have added number of cases and median time interval (median instead of mean, to be consistent with the text of the manuscript) to Figure 1a. Furthermore, we have added the “Before third dose” time point to Figure 1a. The blue lines are the median concentration at all three time points (7.1 at both “After second dose” and “Before third dose” and about 319 at “After third dose”). As we report the median, and not the mean in this figure, we do not think it would be helpful to add a confidence interval to the figure.
  4. We have now added number of patients to Figure 1b. Furthermore, we have changed the ordinate to have the same scale as in Figure 1a. The reason there are no observations above 1000 in Figure 1b, is that this figure shows 95% confidence intervals for the mean (on log scale), while Figure 1a shows all observations as dots. Median time from vaccine to blood collection has been added to Figure 1a, and as this would be the same in Figure 1b, it has not been added here.

Reviewer 2 Report

Manuscript ID: vaccines-1651118 

The authors reported determinants of antibody response to a third SARS-CoV-2 mRNA vaccine dose in solid organ transplant recipients: results from the prospective cohort study COVAC-Tx. 

The results of this manuscript will help to the patients in the hospital and the clinic as brief case report on antibody response to a third SARS-CoV-2 mRNA vaccine dose. 

Author Response

We thank the reviewer for the assessment of the manuscript and the positive evaluation. 

Reviewer 3 Report

This paper by Balsby et al validates results reported by others in the field describing delivery of a third dose of BNT162b mRNA vaccine to patients with solid organ transplants (SOT). The paper makes reference to other similar publications in the field and reports similar trends showing improved humoral responses after the third dose of BNT162b. This paper does report a larger patient study size, and provides more detail on the nature of patients that mounted inferior antibody responses following the third dose (steroid, MMF/aza, older age). The larger cohort and additional multiple regression analyses are added strengths to this particular studies and represent an important contribution to the body of work in this field. The outcomes are also encouraging with few patients developing serious illness despite the less than optimal responses compared to lower risk cohorts of patients studied in other reports. 

We are already administering fourth doses of this class of vaccines to higher risk patients, this clearly points toward the need to shift vaccination strategies for high risk patients. 

Author Response

(The authors gave the same response as above.)

Round 2

Reviewer 1 Report

The suggestions for improvement were taken up by the authors and largely implemented.